# Testing of the therapeutic efficacy and safety of AMPA receptor RNA aptamers in an ALS mouse model

Megumi Akamatsu[1], Takenari Yamashita[1], Sayaka Teramoto[1,2], Zhen Huang[3], Janet Lynch[3], Tatsushi Toda[1], Li Niu[3], Shin Kwak[1,2]

In motor neurons of sporadic amyotrophic lateral sclerosis (ALS) patients, the RNA editing at the glutamine/arginine site of the GluA2 subunit of $\alpha$-amino-3-hydroxy-5-methyl-4-isoxazole propionic acid (AMPA) receptors is defective or incomplete. As a result, AMPA receptors containing the abnormally expressed, unedited isoform of GluA2 are highly $Ca^{2+}$-permeable, and are responsible for mediating abnormal $Ca^{2+}$ influx, thereby triggering motor neuron degeneration and cell death. Thus, blocking the AMPA receptor–mediated, abnormal $Ca^{2+}$ influx is a potential therapeutic strategy for treatment of sporadic ALS. Here, we report a study of the efficacy and safety of two RNA aptamers targeting AMPA receptors on the ALS phenotype of AR2 mice. A 12-wk continuous, intracerebroventricular infusion of aptamers to AR2 mice reduced the progression of motor dysfunction, normalized TDP-43 mislocalization, and prevented death of motor neurons. Our results demonstrate that the use of AMPA receptor aptamers as a novel class of AMPA receptor antagonists is a promising strategy for developing an ALS treatment approach.

## Introduction

Amyotrophic lateral sclerosis (ALS) is the most common adult-onset motor neuron disease, characterized by progressive neurodegeneration of both upper and lower motor neurons (Al-Chalabi & Hardiman, 2013). More than 90% of the ALS patients belong to the sporadic form or do not carry any known ALS-linked genetic mutations (Al-Chalabi & Hardiman, 2013). To date, there are more than 50 ALS-linked genes, including those that encode superoxide dismutase 1 (SOD1), chromosome 9 open reading frame 72 (C9ORF72), transactive response DNA-binding protein 43 (TDP-43) encoding gene (TARDBP), and the fused in sarcoma (FUS) gene (Paez-Colasante et al, 2015). Pathogenesis of ALS has not been well elucidated even in familial ALS cases associated with inheritance of ALS gene mutations. Even though ALS was clinically established more than 150 yr ago, no

therapy has been developed to effectively slow or prevent ALS patients from dying from respiratory muscle failure within several years after disease onset. Currently, there are two FAD-approved ALS drugs (i.e., riluzole and edaravone). None, however, have been effective in improving the quality of life and/or prolonging patients' lives (Kim et al, 2020). As such, there is an unmet, urgent need in developing new and effective drugs for ALS treatment.

The GluA2 subunit of $\alpha$-amino-3-hydroxy-5-methyl-4-isoxazole propionic acid (AMPA) receptors is a potential target for ALS drug development. As one of the four AMPA receptor subunits, GluA2 undergoes a unique RNA editing at the glutamine/arginine (Q/R) site (Seeburg, 2002). This Q/R editing is virtually complete in healthy adult human and mammalian neurons in the central nervous system (CNS) (Seeburg, 2002; Kawahara et al, 2003). However, in the motor neurons of virtually all sporadic ALS patients, the RNA editing at this site in GluA2 is defective or incomplete (Takuma et al, 1999; Kawahara et al, 2004; Kawahara & Kwak, 2005; Kwak & Kawahara, 2005; Hideyama et al, 2012). The RNA editing defect results from the down-regulation of adenosine deaminase acting on RNA 2 (ADAR2), the enzyme that specifically catalyzes the Q/R site editing in GluA2 (Melcher et al, 1996; Gerber et al, 1997; Higuchi et al, 2000). To study the link between deficient ADAR2 expression and motor neuron death, we previously created a conditional ADAR2 knockout mouse line (ADAR2[flox/flox]/VAChT-Cre.Fast; AR2) (Hideyama et al, 2010). AR2 mice exhibit a progressive ALS phenotype (Hideyama et al, 2010; Hideyama & Kwak, 2011), suggesting that RNA editing at the Q/R site of GluA2 is essential for motor neuron health and survival and that the Q/R editing defect in GluA2 is causative to sporadic ALS.

In the vast majority of sporadic ALS patients and some familial ALS patients as well, TDP-43 pathology, namely, the disappearance of TDP-43 from the nucleus and the concurrent formation of the inclusion bodies composed of abnormal aggregated TDP-43 in the cytoplasm, has been observed in lower motor neurons, and hence TDP-43 pathology is considered a pathological hallmark of ALS (Arai et al, 2006; Neumann et al, 2006; Sreedharan et al, 2008; Yokoseki et al, 2008; Suk & Rousseaux, 2020). Motor neurons exhibiting TDP-43 pathology in sporadic ALS patients are invariably devoid of ADAR2 immunoreactivity (Aizawa et al, 2010). In AR2 mice, those

[1]Department of Neurology, Graduate School of Medicine, The University of Tokyo, Tokyo, Japan  [2]Department of Neurology, Tokyo Medical University, Tokyo, Japan  [3]Department of Chemistry, University of Albany, State University of New York, Albany, NY, USA

Correspondence: kwak@tokyo-med.ac.jp; lniu@albany.edu

ADAR2-lacking motor neurons similarly exhibit TDP-43 mislocalization from the physiological localization in the nucleus to cytoplasm (Yamashita et al, 2012). Furthermore, TDP-43 mislocalization can be activated by exaggerated, AMPA receptor-mediated $Ca^{2+}$ influx (Yamashita et al, 2012). In fact, we have shown that in AR2 mice, when abnormal $Ca^{2+}$ influx through AMPA receptors is blocked, TDP-43 mislocalization in the motor neurons of the AR2 mice can be normalized (Yamashita et al, 2012) and consequently the ALS phenotype in AR2 mice can be rescued (Hideyama et al, 2010; Hideyama & Kwak, 2011; Yamashita et al, 2012). These results suggest that abnormal AMPA receptor activity due to the abnormal expression of GluA2Q contributes to TDP-43 pathology in the motor neurons of ALS patients. Therefore, blocking the abnormal AMPA receptor activity should be a viable approach for developing potential ALS drug candidates. In this sense, the AR2 mouse model is a powerful and the ideal platform for testing AMPA receptor inhibitors as potential ALS drug candidates in that the efficacy of rescuing ALS phenotype, along with the safety profile, can be measured. Furthermore, monitoring the normalization of TDP-43 mislocalization in AR2 mice by the use of an AMPA receptor inhibitor should serve as a biomarker for evaluating the therapeutic efficacy of that inhibitor (Akamatsu et al, 2016; Yamashita et al, 2017).

Currently, almost all the AMPA receptor antagonists are small-molecule compounds (Sólyom & Tarnawa, 2002; Grasso et al, 2003). These molecules have low molecular weights and are designed for systemic administration, and by necessity, these compounds are lipophilic to promote intestinal absorption and penetration of the blood-brain barrier (BBB). The water solubility of these compounds is thus limited. We have previously isolated a group of RNA aptamers targeting AMPA receptors (Huang et al, 2010; Huang et al, 2017) using an in vitro evolution process known as systematic evolution of ligands by exponential enrichment (SELEX) (Ellington & Szostak, 1990; Tuerk & Gold, 1990). As antagonists, the RNA aptamers we have isolated show higher potency and selectivity, as compared with traditional, small-molecule compounds, and they are water soluble by nature. We therefore hypothesize that in vivo, the dose of such an aptamer could be administered as low as possible to achieve the therapeutic efficacy with minimal or no adverse effects. Here, we report the first animal study of two chemically modified RNA aptamers in AR2 mice.

## Results

### Preparation of 2′-fluoro (2′-F) modified RNA aptamers for animal testing

In this study, we tested two RNA aptamers, that is, FN1040 and FN58, in AR2 mice. FN1040 is a noncompetitive inhibitor, selecting GluA1 and GluA2 AMPA receptor subunits (Huang et al, 2017). In contrast, FN58 is a competitive antagonist, and it broadly inhibits all three glutamate receptor subtypes, namely, AMPA, kainate, and N-methyl-D-aspartate receptors (NMDA) (FN58 is a 2′-F aptamer that shares the same RNA sequence with its unmodified or regular RNA aptamer AN58, isolated from SELEX) (Huang et al, 2007). These aptamers were assayed for their in vitro potency and selectivity for AMPA receptors using whole-cell recording (Figs 1 and S1) (Huang et al, 2017). As shown (Fig 1B), an aptamer inhibited glutamate-induced

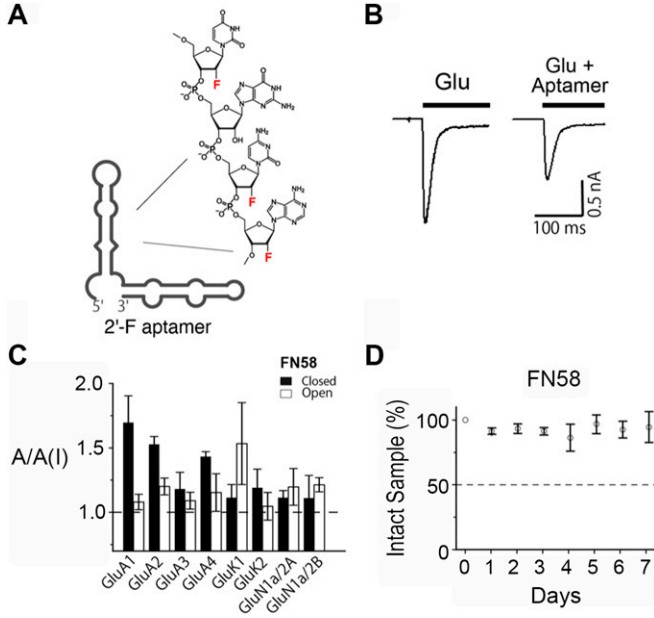

**Figure 1. 2′-F modified RNA aptamers.**
**(A)** An Mfold predicted secondary structure of FN1040 as an example of 2′-fluoro substitution on the ribose sugar. **(B)** A pair of representative whole-cell current responses to glutamate (1 mM) in the absence (left trace) and presence of FN1040 (right trace, 1 $\mu$M FN1040) from an HEK-293 cell that transiently expressed GluA2Q$_{flip}$ AMPA receptors. The bar shows the time course of glutamate exposure to the cell. **(C)** Inhibition of FN58, as an example, on glutamate receptors each of which was transiently expressed in HEK-293 cells. The GluA2 AMPA receptor used for aptamer assay was GluA2Q. For all the assays, 2 $\mu$M of FN58 was tested with two glutamate concentrations (mM), that is, 0.05 and 3 for GluA1, 0.1 and 3 for GluA2, GluA3 and GluA4, 0.05 and 3 for GluK1 and GluK2, and 0.02 and 0.05 for GluN1a/2A and GluN1a/2B, respectively. Low and high glutamate concentrations corresponded to the assay of an aptamer for the closed- and open-channel forms of a receptor (Li & Niu, 2004). The ratio of the whole-cell current amplitude in the absence and presence of an aptamer or A/A(I) is plotted with the SD. For each A/A(I) value, at least three data points were collected from at least three cells. **(D)** In vitro stability of FN58. FN58 was mixed with CSF, and incubated at 37°C for variable lengths of time (days). Samples were visualized on PAGE. The intensity of the ethidium bromide–stained bands was normalized to the control (band from day 0). The average of three sets of samples were used for the plot. The error bars represent SD.

whole-cell current response. By the ratio of the whole-cell current response to glutamate in the absence and presence of FN58, for instance, the aptamer inhibited the activity of virtually all glutamate receptor subunits, although it had a stronger activity on the closed-channel form of the GluA1 AMPA receptor subunit and the open-channel form of the GluK1 kainate receptor activity (Fig 1C and Table S1). Both aptamers were chemically modified in that the 2′-hydroxyl (OH) group was replaced with fluorine (2′-F) in the sugar ring (Fig 1A). This is because unmodified aptamers are cleaved with a $t_{1/2}$ of a few minutes by ribonucleases, the enzymes that catalyze RNA degradation (Brody & Gold, 2000). These enzymes are abundant in cerebrospinal fluid (CSF) (Sawadogo & Roeder, 1985; Pieken et al, 1991). Such a clearance rate makes natural RNA aptamers unsuitable for in vivo use. Substituting 2′-F for 2′-OH group, however, can turn an RNA significantly resistant to ribonucleases with a $t_{1/2}$ of >2 d (Brody & Gold, 2000). For instance, the $t_{1/2}$ of FN1040 is extended to >2 d as compared with just a few minutes for its regular or unmodified counterpart in CSF

(Huang et al, 2017) (Fig S1 and Table S1). FN58 is even more stable in CSF (Fig 1D). In this study, each of the 2'-F modified aptamers was made by enzymatic transcription (see the Materials and Methods section), purified and tested for its antagonistic activity before animal study (Table S1).

### In vivo stability and distribution of [32]P-labeled FN1040

To deliver aptamers to the affected part of the CNS at a concentration that could effectively suppress AMPA receptor activity, we adopted intracerebroventricular administration approach. This delivery route also permitted RNA aptamers to bypass the BBB. Using an osmotic pump, we infused an aptamer to AR2 mice through a brain cannula stereotactically placed in the lateral ventricle (Fig 2A). To examine the distribution and in vivo stability of the infused aptamer, we prepared and used [32]P-labeled FN1040. Autoradiography of cryosections of the forebrain, hindbrain, brainstem, and three segments of spinal cord was examined after 3-d of FN1040 infusion. The results showed that the aptamer was enriched in the gray matter of the brain and the spinal cord (Fig 2B). In addition, the radioactivity in a homogenized tissue sample was quantified using liquid scintillation counting. We found that [32]P-radioactivity was distributed throughout the brain and spinal cord, including gray matter, although the radioactivity was the highest in the forebrain and lowest in the lumbar cord (Fig 2C). This graded distribution was most likely due to a slow diffusion of the aptamer from the site of infusion within the subarachnoid space, including spinal subarachnoid space. To test the putative in vivo stability of the aptamer, we ran SDS–PAGE to analyze the homogenized samples from all tissues after a 72-h infusion. Based on the radioactivity from the [32]P-labeled FN1040 in each of the tissue samples (from forebrain to lumbar cord) (Fig 2D, right panel), we observed that [32]P-labeled FN1040 (Fig 2D, right panel) ran with the same mobility as the non-labeled FN1040 (Fig 2D, left panel). This

result showed that FN1040 was stable in the tissue, as it remained a single, intact band for at least 72 h (Fig 2D, right panel). The in vivo stability for FN1040 was therefore similar to the in vitro stability we reported earlier (Huang et al, 2017).

### Dose-dependent efficacy of FN1040 in AR2 mice by a short-term aptamer administration

Previously, we have shown that mislocalization of TDP-43 is induced by abnormal expression of GluA2Q, which mediates abnormal $Ca^{2+}$ influx (Yamashita et al, 2012); yet perampanel, an AMPA receptor antagonist and a small molecule compound, can normalize the TDP-43 mislocalization (indicative of ameliorated $Ca^{2+}$ influx) and prevent the ALS phenotype of AR2 mice (Akamatsu et al, 2016). Thus, we decided to choose perampanel as a control for the test of whether an aptamer could similarly normalize TDP-43 mislocalization and rescue motor neurons.

First, we monitored four groups of AR2 mice for behavioral changes at different time points. Two mouse groups received either FN1040 (30 $\mu$M) or FN58 (20 $\mu$M), and the other two received vehicle and peroral administration of perampanel, respectively. All the mice except in the group that received perampanel were subject to isoflurane anesthesia for surgery required for commencing intracerebroventricular delivery of aptamer. Specifically, we measured the duration of move-and-walk (Fig 3A, left panel) and the time of standing by hindlimbs (rearing) (Fig 3A, right panel) in 3 min at each time point. All the mice in the group that received perampanel (n = 5) were administered with 20 mg/kg, a dose that effectively rescued ALS phenotype of AR2 mice as we determined from our previous study (Akamatsu et al, 2016). These mice, however, were akinetic or moved slowly, if ever, after 30 min and fell completely into deep sleep 60 min after the drug administration (Fig 3A, purple line). Even 24 h after perampanel administration, their physical activities did

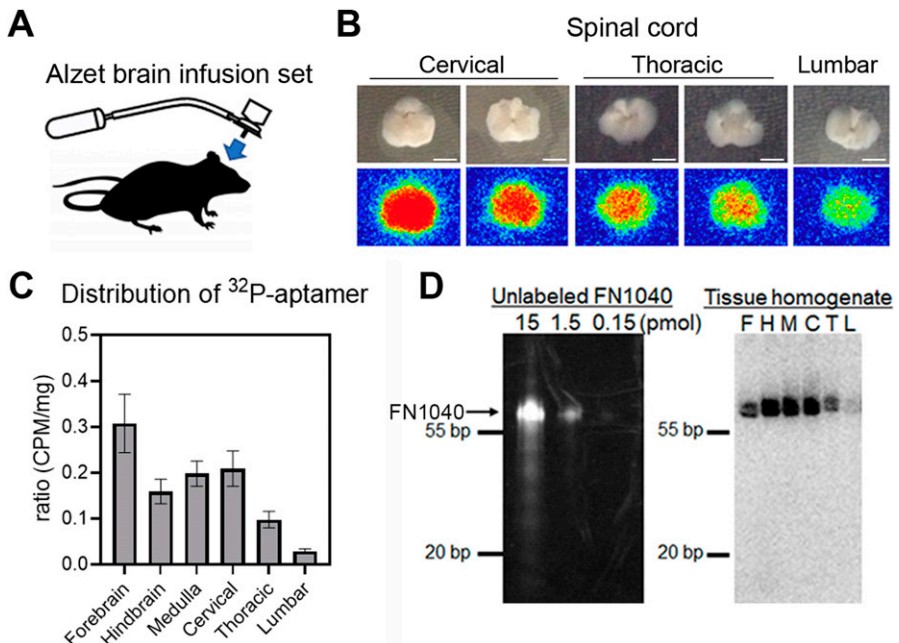

**Figure 2. Distribution of [32]P-labeled FN1040 in the CNS after continuous cerebroventricular administration.**
**(A)** Schema of the brain infusion assembly. An Alzet osmotic pump type 2002 was connected to an indwelling brain infusion cannula with a catheter, which enabled continuous infusion of the aptamer contained in the pump into the lateral cerebroventricle at a flow rate of 1 $\mu$l/h for 7 d. **(B)** Axial sections of the spinal cord (upper row) and the autographic image of corresponding sections (lower row) from the mice infused with [32]P-labeled RNA aptamer for 72 h. The autoradiographic imaging was analyzed by the FUJIFILM Bio-imaging Analyzer BAS-2500. Scale bar = 2 mm. **(C)** Radioactivity of different brain area analyzed with liquid scintillation counter. Vertical axis is the ratio of counts per minute (CPM) per milligram in each area to total CPM per milligram of four individual experiments. Mean ± SEM. **(D)** SDS–PAGE analysis of FN1040. Left: Different amounts of unlabeled FN1040 (arrow). Right: autoradiography of SDS–PAGE of tissue homogenate of a mouse infused with [32]P-labeled FN1040 after treatment with proteinase K. F, forebrain; H, hindbrain; M, medulla oblongata; C, cervical cord; T, thoracic cord; L, lumbar cord.
Source data are available online for this figure.

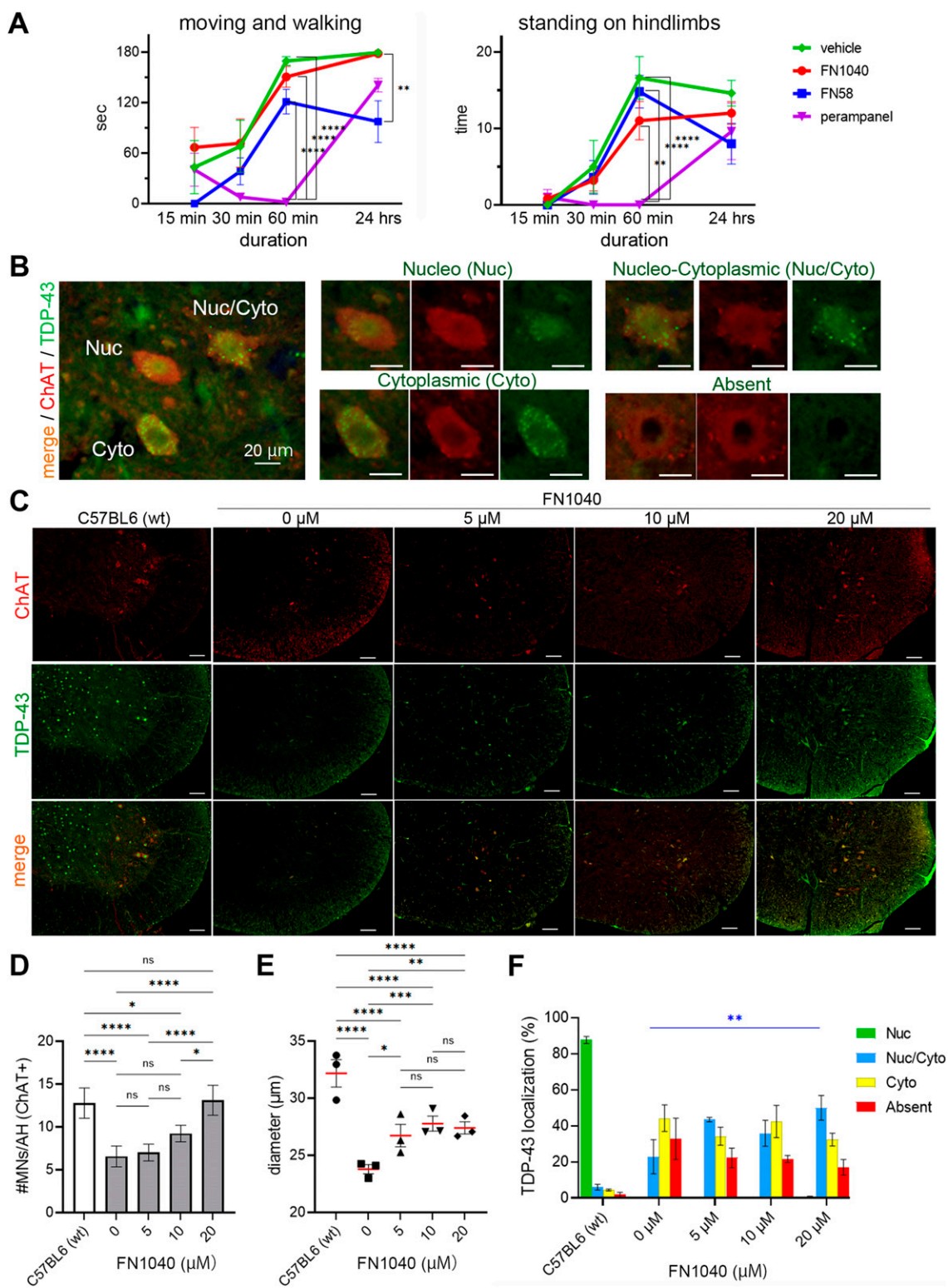

**Figure 3. Effects of short-term administration of FN1040 on AR2 mice.**
**(A)** Behavioral changes. FN1040, FN58, or vehicle was intracerebroventricularly infused through surgically implanted cannula under isoflurane anesthesia and Food and Drug Administration-approved α-amino-3-hydroxy-5-methyl-4-isoxazole propionic acid antagonist perampanel was administered orally. AR2 mice infused with FN1040 at a concentration of 30 μM (n = 5; red) exhibited active locomotion (moving and walking) in the same level as vehicle-treated AR2 mice (n = 5; green) during 24 h after surgery. In contrast, mice receiving perampanel at an effective dose (20 mg/kg, n = 5; purple) exhibited marked inactivity. AR2 mice infused with FN58 at a concentration of 20 μM (n = 5, blue) exhibited the delay in recovery compared with FN1040, and the sedation prolonged to the following day. Left: total time mice moved and walked

not recover to the level of vehicle-treated mice (Fig 3A, green line). The physical activity of these mice only returned normal 2 wk after the last drug administration. In contrast, the AR2 mice infused continuously with FN1040 at 30 µM exhibited similar physical activity throughout aptamer administration, as compared with the vehicle control (Fig 3A, red versus green line). It is noteworthy that a continuous infusion of aptamer FN1040 at 30 µM effectively reduced $Ca^{2+}$ influx through AMPA receptors (see the section below) but did not induce sedation, unlike perampanel. On the other hand, the AR2 mice that received infusion of 20 µM FN58 exhibited slow recovery in moving and walking, and their physical activities remained low on the following day (Fig 3A, blue line). These results suggested that a continuous infusion of aptamer FN1040, even at a higher concentration (i.e., 30 µM), was safe or without causing appreciable sedative effect on mice, as compared with the vehicle control and with either perampanel or FN58. Based on this initial safety assessment of the two aptamers, we decided to focus on FN1040 for the rest of the studies.

Next, we carried out a histological study of the spinal cord tissue to examine whether FN1040 was efficacious in normalizing TDP-43 mislocalization (Figs 3B and S2A) and therefore rescuing motor neurons (Saraiva et al, 2016). After a 14-d aptamer administration, the number of choline acetyltransferase (ChAT)-positive motor neurons in the spinal cord was significantly higher in mice that received FN1040 than those that received just artificial CSF (i.e., vehicle control) (Fig 3C and D). In fact, the increase of the number of ChAT-positive motor neurons was dose-dependent, and the increase was the highest in mice treated with 20 µM of FN1040 (Figs 3D and S2B). The size of ChAT-positive neurons in mice that received FN1040 was also larger as compared with untreated or vehicle-treated mice (Figs 3E and S2C). The percentage of the motor neurons exhibiting TDP-43 immunoreactivity in both the nucleus and the cytoplasm increased as well (i.e., the blue bar in Fig 3F), whereas the percentage of the motor neurons lacking TDP-43 immunoreactivity decreased in mice that received FN1040, as compared with the vehicle-treated mice (Figs 3F and S2D). These results showed that FN1040 effectively reduced the TDP-43 cytoplasmic mislocalization and thus rescued motor neurons from neurodegeneration even in a short-term (14 d) treatment.

To compare the efficacy of FN1040 with not just the vehicle control, we also intracerebroventricularly infused FN58 at 20 µM concentration for a 14-d period. Unlike FN1040 treatment, infusion of FN58 caused locomotion reduction and sleepiness, although the AR2 mice did tolerate the intracerebroventricular infusion. Spinal motor neurons were similarly examined after 14-d infusion. We observed that the number (Fig S3A and B) and size (Fig S3C and D) of

motor neurons became higher and larger, respectively, and the TDP-43 subcellular localization was also improved (Fig S3E). Thus, FN58 appeared no less efficacious than FN1040 (Fig S3). Taken together, our results demonstrated that aptamer-treated mice had more healthy motor neurons than those vehicle-treated mice, and the rescue of motor neurons was dose-dependent as well. Our results further suggested that a concentration not lower than 20 µM for FN1040 would be preferred.

## Long-term administration of FN1040 in AR2 mice rescued motor neurons and improved motor function

Based on the short-term safety and efficacy study (Fig 3), we conducted a long-term (12-wk) administration of FN1040, along with FN58, in AR2 mice to evaluate long-term efficacy on health and survival of motor neurons by measuring the motor function and morphological changes in motor neurons of aptamer-treated and -untreated AR2 mice. We found that all the AR2 mice treated with 30 µM FN1040 behaved normal during aptamer infusion, whereas the mice treated with FN58 at the same concentration, that is, 30 µM, exhibited severe sedative behavior. We, therefore, decided to proceed the long-term study with the 30 µM concentration of FN1040 but only 20 µM concentration of FN58 to minimize the side effect. However, even when the concentration of FN58 was lowered to 20 µM, some mice that received FN58 still became hypoactive but only during the first few days of infusion. As such, mice were treated with 5 µM FN58 in the first 3 d and then the concentration of FN58 was escalated to 20 µM. We found mice tolerated such a treatment regimen during the 12-wk experiment.

We first performed rotarod test for measuring motor function. We selected a group of 6-wk-old AR2 mice and trained them to remain on rotarod for the long-term assessment. Once trained, those AR2 mice were divided into three subgroups (n = 13/group). We started the treatment of AR2 mice at 20 wk of age when the motor neuron degeneration and motor dysfunction were clearly under way (Hideyama et al, 2010). A 20-wk age of AR2 mice was still considered an early phase of the disease progression. Unlike other ALS model mice such as SOD1 transgenic mice, the disease progression of AR2 mice is slow and represents more closely the time course of human ALS (Hideyama et al, 2010). As such, AR2 mice at 20–32 wk of age used in our experiments roughly represented the disease progression from the early to middle stages.

Before aptamer or vehicle treatment, no significant difference was observed among the three subgroups for either the initial rotarod retention time (Fig 4A) or latency to fall (Fig 4B) or body weight (Fig 4C). However, the rotarod retention time was markedly

during 180 s of observation. Right: the number of rearing (standing on hindlimbs) during 180 s of observation. Mean ± SEM. Two-way ANOVA followed by Tukey's multiple-comparisons test. **$P < 0.01$, ****$P < 0.0001$ (Tables S2 and S3). **(B)** Representative image of TDP-43 immunohistochemistry (green: anti-TDP-43 antibody positive area). TDP-43 localized within the nucleus of motor neurons (Nuc) of normal mice, whereas it localized in the cytoplasm (Nuc/Cyto and Cyto) or disappeared (Absent) in AR2 mice. Scale bar = 20 µm. **(C)** Anterior horns of the cervical spinal cord double stained with anti-ChAT (red) and anti-TDP-43 (green). Scale bar = 100 µm. **(D)** The number of ChAT-positive cells in the anterior horn of the cervical spinal cord was significantly larger in the 20 µM of FN1040 group compared to that in the control (artificial CSF) group and lower FN1040 concentration groups. Six anterior horn cells per mouse, three mice per group. **(E)** Mice treated with FN1040 showed a dose-dependent increase in the diameter of ChAT-positive cells. **(F)** The proportion of ChAT-positive cells with nuclear localization (Nuc/Cyto + Nuc) was higher in the FN1040-treated mice compared to that in vehicle-treated mice (n = 3 in each group). Nuc, nucleo; Nuc/Cyto, nucleo-cytoplasmic; Cyto, cytoplasmic; Absent, lack of TDP-43 immunoreactivity. *$P < 0.05$, **$P < 0.01$, ***$P < 0.001$, ****$P < 0.0001$. Mean ± SEM, two-way ANOVA followed by Tukey's multiple-comparisons test (Table S4).
Source data are available online for this figure.

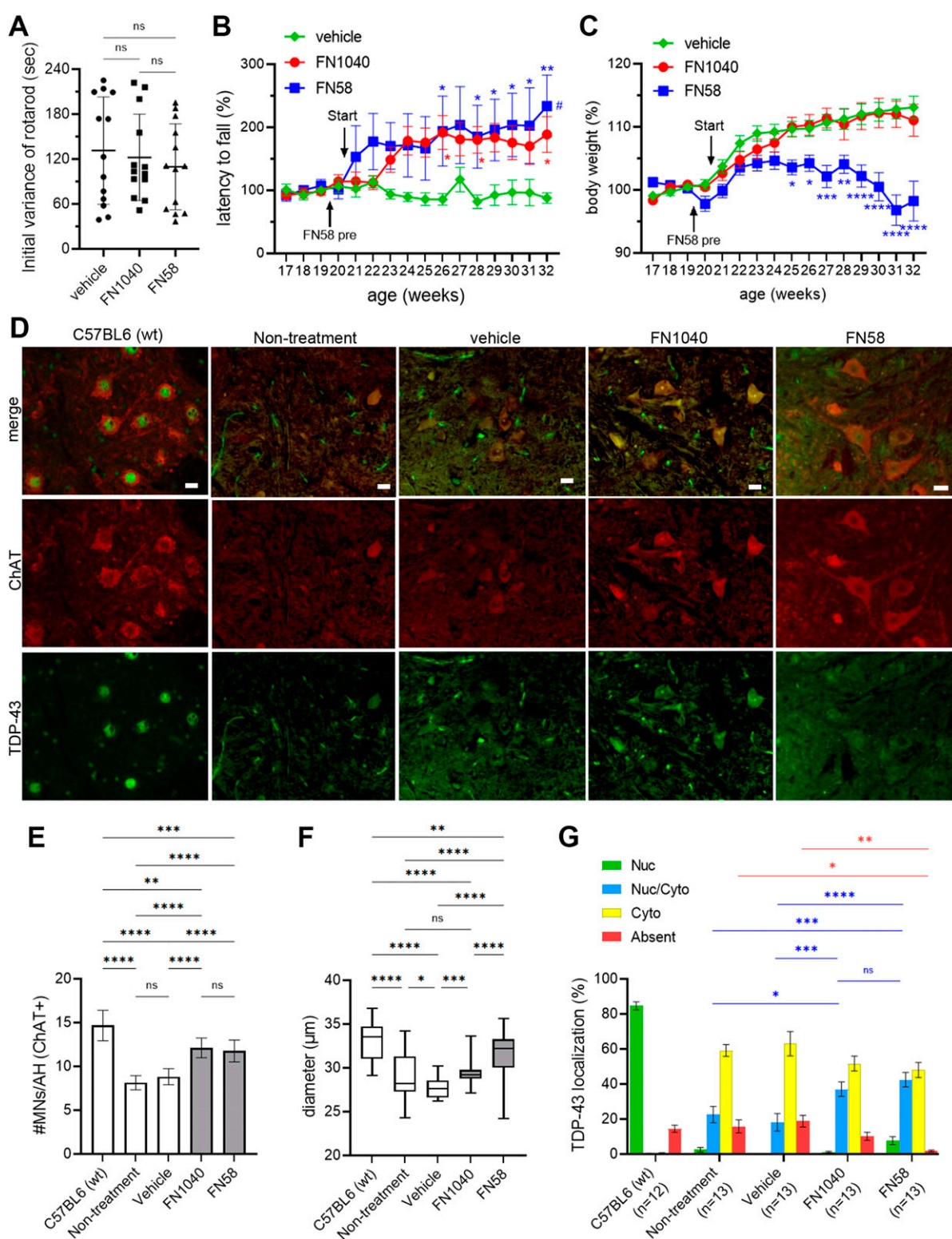

**Figure 4. Rescue of motor function and death of motor neurons after long-term administration of FN1040 and FN58.**
**(A)** Initial rotarod scores were not significantly different among the FN1040-treated (n = 13), the FN58-treated (n = 13), and the vehicle-treated (n = 13) AR2 mice. Mean ± SEM, one-way ANOVA followed by Tukey's multiple-comparisons test. **(B)** Mice received either FN1040 or FN58 were able to retain on the rotarod longer than the vehicle-treated mice. FN58-pre: administration of initial dose of FN58 (5 $\mu$M). Mean ± SEM, two-way ANOVA followed by Tukey's multiple-comparisons test, *$P < 0.05$, **$P < 0.01$ versus vehicle. #$P < 0.05$ versus initial value (Table S5). **(C)** Mice treated with FN1040 exhibited body weight change similar to those treated with vehicle, whereas those treated with FN58 failed to increase the body weight 1 mo after administration and thereafter. Mean ± SEM, two-way ANOVA followed by Tukey's multiple-comparisons

and steadily improved just after a few weeks of administration of either FN1040 or FN58, as compared with the rotarod score in either the pre-experimental period for the same aptamer group or the vehicle group (Fig 4B). The improved rotarod performance was also maintained throughout the testing period in both aptamer groups, as compared with their pretesting levels (Figs 4B and S4A). In contrast, the rotarod performance of the AR2 mice that received vehicle treatment remained the same, evidenced by a virtually flat rotarod score before and after the vehicle injection or throughout the entire testing period (Figs 4B and S4A). FN1040 had no adverse effect on either body weight or growth during this 12-wk testing period, whereas FN58-infused AR2 mice failed to gain body weight (Fig 4C). In fact, AR2 mice exhibited body weight gain over time, which was similar to control mice until about the age of 1 yr (Hideyama et al, 2010). No overt disabilities were observed in either FN1040-treated or FN58-treated mice. These results showed that aptamer treatment improved the motor function and such an improvement was maintained during the treatment. Clearly, FN1040 was a better aptamer by safety, as compared with FN58, although both aptamers were almost equally efficacious on improving motor function. Furthermore, after a 12-wk infusion, we found the number and the size of ChAT-positive motor neurons were significantly higher and larger in both FN1040- and FN58-treated groups, as compared to either the vehicle-infused or the untreated group (Fig 4D–F). Such an improvement in the motor neuron health was consistent with the short-term efficacy. In addition, the percentage of the ChAT-positive motor neurons with the normal, nuclear TDP-43 localization (Nuc and Nuc/Cyto) was significantly higher in the aptamer-treated groups, as compared to either vehicle-treated or untreated groups (Figs 4G and S4B and C). Although the improvement in normalizing TDP-43 mislocalization appeared to be subtle, as compared with the vehicle-treated mice, the rotarod functional test score showed a remarkable improvement, suggesting a significant proportion of motor neurons recovered their neuronal function after the aptamer treatment.

The results from the long-term study demonstrated that FN1040- and FN58-treated AR2 mice exhibited a significant improvement in ALS phenotypes and such a behavioral improvement was correlated to the increase in both the number and the morphological change in motor neurons. These positive changes observed in the long-term treatment (Fig 4) were also consistent with those seen in the short-term infusion (Fig 3). We therefore concluded that the use of FN1040 achieved the therapeutic efficacy and maintained it safely throughout aptamer treatment regimens. Because the histological parameters between the non-treatment and the vehicle treatment groups were similar, the aptamer delivery approach, that is, intracerebroventricular delivery of our aptamers, throughout the entire infusion period, was safe as well.

## Discussion

The current study has demonstrated RNA aptamers acting on AMPA receptors are capable of blocking exaggerated $Ca^{2+}$ influx through abnormal AMPA receptors containing the GluA2Q isoform in AR2 mice, where motor neurons undergo neurodegeneration as in sporadic ALS. The use of $^{32}P$-labeled FN1040 enabled us to assess and confirm the in vivo stability of a chemically modified aptamer in the CSF of living mice and visualize a widespread distribution of the aptamer throughout the brain and spinal cord. By intracerebroventricular infusion for just 2 wk, either FN1040 or FN58 is already capable of normalizing the TDP-43 mislocalization and increasing the number and the size of motor neurons. A 12-wk continuous infusion of either FN1040 or FN58 further extends the improvement of the motor performance of these AR2 mice. Our study further shows that the use of FN1040 is without adverse effects, such as sedation.

When both the efficacy and the safety of FN1040 is compared with FN58, we may draw some mechanistic implications. Both FN58 and FN1040 are efficacious in rescuing AR2 mouse phenotypes but FN58 generates some sedative effects, whereas FN1040 does not. Mechanistically, FN1040 is a noncompetitive inhibitor selective to AMPA receptors, and it binds to a site distinct to the glutamate binding site. In contrast, FN58 is a competitive inhibitor (Huang et al, 2007), and its broad activity against all three glutamate receptor subtypes may have contributed to its sedative effect. Thus, this study suggests that targeting the right, noncompetitive site on AMPA receptors may offer an effective means to block the abnormal, receptor-mediated $Ca^{2+}$ entry with minimal or no adverse effects. In fact, administration of FN1040 has not caused any adverse changes on mouse behaviors and is well tolerated throughout a 12-wk continuous aptamer infusion even at a high dosage (i.e., 30 $\mu$M). This is important from the clinical viewpoint, as adverse reactions, such as CNS suppression, to the administration of an AMPA receptor inhibitor drug candidate are a general hurdle to a successful drug development.

We have shown administration of perampanel effectively improves ALS phenotype but generates sedation in the AR2 mice although the tolerance can be developed (Akamatsu et al, 2016). In this study, FN58 similarly causes sedation but the tolerance can be similarly developed and managed as well if an infusion concentration is lowered initially. In addition, perampanel is known to elicit a strong dose-dependent motor impairment in mice in a rotarod

---

test, *$P < 0.05$, **$P < 0.01$, ***$P < 0.001$, ****$P < 0.0001$ versus vehicle (Table S6). **(D, E, F, G)** show morphological change. **(D)** Anterior horns of the cervical spinal cord double stained with anti-ChAT (red) and anti-TDP-43 (green). Scale bar = 20 $\mu$m. **(E, F)** show the number and the size of ChAT-positive cells, respectively, in the anterior horn of the cervical spinal cord from FN1040- or FN58-treated AR2 mice (n = 13), as compared with those in the untreated control AR2 mice (n = 13), vehicle-treated AR2 mice (n = 13) or C57BL4 (wt) mice (n = 12). 10 anterior horn cells per mouse were used for cell counting. Mean ± SEM, two-way ANOVA followed by Tukey's multiple-comparisons test. ns, not significant, *$P < 0.05$, **$P < 0.01$, ***$P < 0.001$, ****$P < 0.0001$. **(G)** The proportion of the four TDP-43 localization patters in the ChAT-positive cells. Mean ± SEM, two-way ANOVA followed by Tukey's multiple-comparisons test, compared with the nontreated mice (bars and asterisks in blue color and in red color are for the comparison for Nuc/Cyto and absent, respectively) (Table S7). Total number of MNs used for TDP-43 localization analysis; C57BL6 (n = 1,702), nontreated (n = 1,055), vehicle (n = 1,146), FN1040 (n = 1,575), and FN58 (n = 1,529).
Source data are available online for this figure.

test at high doses (Hanada et al, 2011), and a significant percentage of epilepsy patients who take perampanel as an anti-epileptic drug actually experience dose-dependent side effects (Greenwood & Valdes, 2016). In comparison with perampanel, FN1040 is water soluble and has a higher selectivity towards the GluA1 and GluA2 AMPA receptor subunits (Huang et al, 2017; Fukushima et al, 2020). It is conceivable that the lack of adverse behavioral change in mice with FN1040 is likely due to a combination of its noncompetitive nature as an inhibitor with a higher selectivity towards the GluA1 and GluA2 AMPA receptor subunits (Huang et al, 2017) and its unique molecular property as an RNA molecule. The results from this study have shown that developing mechanism-based AMPA receptor RNA aptamers seems to be a promising therapeutic approach.

RNA aptamers cannot go through BBB, due to their sizes and charges. As we have demonstrated in this study, the use of intracerebroventricular infusion has enabled an aptamer to circumvent the BBB and directly reach the CNS. Although this invasive delivery route is less desirable, such an approach may achieve a greater drug concentration in CNS, while possibly minimizing systemic toxicity. On the other hand, developing a noninvasive route of administration for RNA aptamers is possible. For instance, the use of nanoparticle formulations to encapsulate drug molecules for CNS delivery has been explored for several neurodegenerative diseases (Saraiva et al, 2016; Anraku et al, 2017).

The results from the present study have demonstrated that using aptamers to block elevated AMPA receptor activities mediated through abnormally expressed, $Ca^{2+}$ permeable AMPA receptors that contain GluA2Q is the basis of motor neuron rescue, and thus not surprisingly, aptamer treatment is able to improve the motor function of AR2 mice. These results should allow us to draw two major implications. First, recent evidence suggests that a reduction of ADAR2 expression has been observed in the motor neurons of ALS patients who carry ALS-linked gene mutations, such as $FUS^{P525L}$ mutation (Hideyama et al, 2012; Aizawa et al, 2016). Furthermore, ADAR2 is thought to be mislocalized from the nucleus to the cytoplasm in motor neurons and also in human induced pluripotent stem cell-derived motor neurons from ALS patients carrying the C9ORF72 gene with hexanucleotide repeat expansion (Moore et al, 2019). Based on our results and the reports from others, cited above, AMPA receptor antagonists could be potentially effective in not only sporadic ALS but also some other forms of familial ALS. Second, the present study also confirms our previous finding that subcellular localization of TDP-43 is a reliable biomarker for measuring the therapeutic efficacy of AMPA receptor antagonists (Akamatsu et al, 2016; Yamashita et al, 2017). In this context, size shrinkage and the reduction of the number of the motor neurons could be an important pathological benchmark of ALS (Kiernan & Hudson, 1993; Kanazawa, 2001). That the use of FN1040 leads to increase in both the number and the size of motor neurons, which is correlated to the improvement of motor performance, lends a strong support of the therapeutic potential of FN1040. It should be noted that even an infusion of FN1040 for just 2 wk is able to normalize TDP-43 mislocalization, thereby rescuing the motor neurons.

In summary, this study has shown that RNA-based AMPA receptor antagonists or RNA aptamers capable of blocking the AMPA receptor-mediated $Ca^{2+}$ influx are potentially useful as a new class of ALS drug candidates. FN1040 is stable in vivo, well tolerated, and efficacious in rescuing TDP-43 mislocalization and motor neurons, thus increasing motor neuron health and improving motor function of the AR2 mice. Even at a high concentration and a continuous infusion, FN1040 generates a robust efficacy but without any appreciable adverse effects. Our data indicate that using AMPA receptor aptamers through subarachnoidal route provides a promising approach for developing a new ALS treatment option. This study further opens the possibility of even lowering the dose and/or using intermittent instead of continuous infusion to administer the aptamer for achieving and maintaining efficacy.

# Materials and Methods

### Study design

The aim of the study was to evaluate the efficacy and safety of AMPA receptor-selective RNA aptamer FN1040 and FN58 in the ADAR2$^{flox/flox}$/VAChT-Cre.Fast (AR2) mice, a model of sporadic ALS (Hideyama et al, 2010; Hideyama & Kwak, 2011). After establishing the method that enabled continuous infusion of FN1040 into the mouse cerebral ventricle, we examined $^{32}$P-labeled FN1040 for the distribution in the brain and spinal cord and the stability in vivo and confirmed that full-length or intact FN1040 was distributed in the gray matter of throughout the brain and the spinal cord with a rostro-caudally decreasing gradient. After confirming the safety of FN1040 administration via intracerebroventricular injection, we searched for the optimum tolerable doses by 2-wk-administration experiment. Based on the short-term administration experiment, we performed 12-wk long-term infusion experiment. We evaluated the efficacy by measuring changes in the motor function, the number of motor neurons in the spinal cord and immunohistochemical subcellular localization of TDP-43 mislocalization in the motor neurons compared to untreated AR2 mice.

### Mouse models

All animal studies were approved by the Committee on Animal Handling at the University of Tokyo (approval No. Med-P18-008) and were performed in accordance with the guidelines for animal experiments of both the Japan Ministry of Education, Culture, Sports, Science and Technology and the National Intistute of Health (NIH) at the United States. The mice were housed at two to five per cage on a 12-h/12-h light/dark cycle, 23°C with free access to food and water, and with some enrichments.

### AR2 mice

20–32-wk-old male Homozygous (ADAR2$^{flox/flox}$/VAChT.Cre-Fast; AR2) conditional ADAR2 knockout mice via Cre-loxP system were used in this study (Hideyama et al, 2010; Hideyama & Kwak, 2011). In these mice, Cre is selectively expressed in motor neurons under the control of the vesicular acetylcholine transporter promotor (Misawa et al, 2003), ablating the ADAR2$^{flox}$ gene in ~50% of motor neurons by the age of five postnatal wk. As a result, 100% of GluA2 is

unedited at the Q/R site in the motor neurons of AR2 mice. In the AR2 mice, the expression of GluA2 that is unedited at the Q/R site results in the slowly progressive death of motor neurons via $Ca^{2+}$-permeable AMPA receptor-mediated mechanisms (Hideyama et al, 2010; Hideyama & Kwak, 2011), and AR2 mice undergo slow and progressive decline of motor function like as ALS patients. The number of the mice used in all the experiments was based on the power analysis with $\alpha$ and confidence being 0.8 and 0.05, respectively.

### Preparation of 2′-fluoro RNA aptamers for in vivo study

The aptamers used in this study were previously discovered using SELEX (Huang et al, 2007, 2017). A 2′-F–modified RNA aptamer was prepared by transcribing the corresponding DNA template. To synthesize 2′-F RNA aptamer from its DNA template, we incorporated non-canonical 2′-F-NTP (i.e., "N" in the "NTP" stands for adenosine [citidine, uracil] triphosphate ATP, CTP, and UTP), using a mutant T7 RNA polymerase. However, all G (guanosine) positions contained the regular, unmodified G (Huang et al, 2007, 2017). A transcribed RNA was purified using a Bio-Rad Prep Cell (model 491), a PAGE-based, continuous elution apparatus, coupled with a Q column (Huang et al, 2007, 2013). A purified RNA aptamer was tested for its biological activity against its target using whole-cell recording (see Fig 1 and legend), and the procedure of whole-cell recording was previously described (Huang et al, 2013).

### Aptamer infusion

Aptamer FN1040 (Huang et al, 2017) or FN58 (Table S1) was dissolved in artificial CSF containing 100 U/ml of heparin sodium (#9041-08-1; Wako) and 3 $\mu$M-transfer RNA (phenyl-alanine specific from brewer's yeast, R4018; Sigma-Aldrich). Then the solution was filled in an Alzet osmotic pump which had been incubated in the sterile saline at 37°C, 12 h before implantation. The osmotic pump was connected to the brain infusion cannula (Brain Infusion Kit 3; Alzet) with a Teflon tube, thereby enabling continuous delivery of FN1040 or FN58 solution into the mouse lateral ventricle for 3 d (type 1003) or 14 d (type 2002).

### Surgical implantation

Under deep anesthesia with isoflurane, the mouse was situated in the stereotaxic apparatus (Narishige), and the brain infusion cannula (Brain Infusion Kit 3; Alzet) was stereotactically placed at A/P = −0.5 mm, M/L = 1.1 mm, 2.5 mm depth from bregma, followed by fixation to the skull with dental cement (LUTING Versa; GC). The infusion cannula was connected to the aptamer-containing Alzet osmotic pump with a Teflon tube. The osmotic pump, the infusion cannula and the Teflon tube were implanted subcutaneously. For the long-term administration, the Alzet pump type 2002 was changed every 2 wk under anesthesia.

### Distribution and stability

[γ³²P]-ATP (NEG002Z250UC; ParkinElmer) was labeled to RNA aptamer FN1040 using KinaseMax (AM1520; Thermo Fisher Scientific). After continuous infusion of 1.5 $\mu$M of radiolabeled FN1040 for 3 d at a rate of 1 $\mu$l/h, brain and spinal cord were removed under deep anesthesia. The forebrain, hindbrain, medulla, cervical cord, thoracic cord, and lumbar cord were sectioned. After homogenization in the PBS (same v/w for each tissue), radioactivity was mesured in a liquid scintillation counter (Tricarb 3110TR; PerkinElmer). Radioactivity was expressed as the effective CPM value per milligram of tissue weight. Separate sections of 1 mm-thickness of each brain area were subjected to autoradiography (BAS-2500; GF Healthcare). For examining the stability of RNA aptamer in vivo, aliquot of the tissue homogenates was analyzed with SDS–PAGE (TBE-Urea Gels, 10% PAGE, 7 M-Urea #100031466; Invitrogen) after treatment with proteinase K (High Pure Template Preparation Kit, #11796828001; Roche) at 55°C for 3 h. Unlabeled full-length RNA aptamer FN1040 was used as the reference.

### Short-term and long-term administration via cerebroventricle

An aptamer was infused into the lateral ventricle by the method described above. Alzet pump type 2002 was used. For short-term administration, mice were infused with 5, 10, or 20 $\mu$M of FN1040, or 20 $\mu$M of FN58 for 2 wk. For long-term administration, 30 $\mu$M of FN1040 or 20 $\mu$M of FN58, predicted optimum doses for each aptamer, was continuously infused for 12 wk by changing an Alzet pump every 2 wk under anesthesia. For FN58, because of sedative effects observed at 20 $\mu$M concentration, mice were received tolerable concentration of 5 $\mu$M FN58 at a rate of 1.0 $\mu$l/h (dose/h is 1/2 of the dose for 2-wk testing) for 3 d using 1003 Alzet pump before infusion of 20 $\mu$M FN58 using 2002 Alzet pump.

### Behavioral change

Sedative effects were measured by the duration of spontaneous movement and the number of rearing during 3 min period at 15, 30, and 60 min, and 24 h after anesthesia during implantation surgery of cannula in the aptamer-treated or vehicle-treated mice. To compare the sedative effects among the AMPA receptor antagonists, mice receiving oral dose of 20 mg/kg of Food and Drug Administration-approved AMPA receptor antagonist perampanel were observed in the similar method. Behavioral changes were analyzed by the duration time remaining on the rotarod. The mice were placed on a rotarod (MK-610A; Muromachi Kikai Co. LTD), the speed of which linearly accelerated from 4 to 40 rpm over 240 s. The maximum latency to fall from the rod out of three runs was recorded. The mice used in this study were well trained on the rotarod task once a week after the sixth week of age before the infusion experiment. We used 20-wk old male AR2 mice for the functional tests. All the mice in the long-term infusion experiment were tested for the latency to fall from rotarod once a week and monitored the sedative symptoms for 12 wk.

### Immunohistochemistry

After a pre-determined administration period, mice were euthanized under deep isoflurane anesthesia and brains and spinal cords were immediately removed and fixed with 3.5% paraformaldehyde (P0018; Tokyo Chemical Industry), 0.5% glutaraldehyde (G004; TAAB), in PBS at 4°C for 36 h. Spinal cord sections (10 $\mu$m

thickness) were immunostained with anti-TDP-43 polyclonal antibody (10782-2-AP, 1:200; Proteintech) and with anti-ChAT polyclonal antibody (AB144P, 1:200; Millipore) followed by anti-rabbit IgG antibody Alexa 488 (A21441, 1:200; Molecular probes) and anti-goat IgG antibody Alexa 555 (A21432, 1:200; Molecular probe), respectively. Sections were observed under BIOREVO BZ-9000 and BZ-X800 microscope (Keyence), and the number of cells with a diameter of ≥20 $\mu$m in AHCs was quantified in six anterior horn per animal for 2-wk treatment and in 10 anterior horn per animal for long-term treatment using ImageJ software (NIH) and BZ-II analyzing application (Keyence). The immunoreactive signal intensity was analyzed with ImageJ software, with a grayscale background intensity of less than 20 (11.6 ± 0.55; mean ± SEM, max 19.9 gray). TDP-43 positivity was defined as an intensity threefold greater (≥60 gray) than that of the background intensity. The TDP-43 distribution pattern was classified as predominantly nuclear (N), cytoplasmic (C), or nucleocytoplasmic (N/C) as previously reported (Akamatsu et al, 2016).

## Supplementary Information

## Acknowledgements

This work was supported by Japan Society for the Promotion of Science (JSPS) KAKENHI grants JP26640036 (S Kwak) and JP17K09747 (M Akamatsu), Ice Bucket Challenge (IBC) grant and grant from Japan ALS association (S Kwak and M Akamatsu), and National Institute of Health (NIH) grant R21 NS106392-01 (L Niu and S Kwak). We express our gratitude to Prof. Kazunori Kataoka (Tokyo University) for providing us with research facilities and valuable discussions, Prof. Takeshi Iwatsubo (Tokyo University) for providing us with research facilities, Prof. Hitoshi Okazawa and Drs. Kazuhiko Tagawa and Xigui Chen (Tokyo Medical and Dental University) for valuable discussions, Dr. Atsushi Enomoto (Tokyo University) for advising radioisotope experiments. We thank Dr. Takashi Hosaka, Dr. Naoki Hirose, and Ms. Keiko Izumi for technical assistance.

## Author Contributions

M Akamatsu: conceptualization, data curation, formal analysis, funding acquisition, validation, visualization, methodology, and writing—original draft, review, and editing.
T Yamashita: conceptualization, data curation, supervision, funding acquisition, validation, methodology, project administration, and writing—original draft, review, and editing.
S Teramoto: data curation, formal analysis, validation, methodology, and writing—review and editing.
Z Huang: conceptualization, resources, data curation, formal analysis, validation, visualization, methodology, and writing—review and editing.
J Lynch: resources, data curation, formal analysis, validation, visualization, methodology, and writing—review and editing.
T Toda: resources, data curation, formal analysis, funding acquisition, validation, project administration, and writing—review and editing.
L Niu: conceptualization, resources, formal analysis, funding acquisition, methodology, project administration, and writing—review and editing.
S Kwak: conceptualization, resources, data curation, formal analysis, supervision, funding acquisition, validation, methodology, project administration, and writing—original draft, review, and editing.

## Conflict of Interest Statement

The authors declare that they have no conflict of interest.

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
