## [Reviewer comments · Life Science Alliance]

Life Science Alliance

Testing of the therapeutic efficacy and safety of AMPA receptor RNA aptamers in an ALS mouse model

Megumi Akamatsu, Takenari Yamashita, Sayaka Teramoto, Zhen Huang, Janet Lynch, Tatsushi Toda, Li Niu and Shin Kwak

DOI: <https://doi.org/10.26508/lsa.202101193>

Corresponding author(s): *Dr. Shin Kwak (Tokyo Medical University)*

Review Timeline:

Submission Date:	2021-08-14
Editorial Decision:	2021-09-03
Revision Received:	2021-11-26
Editorial Decision:	2021-12-15
Revision Received:	2021-12-23
Accepted:	2021-12-23

Scientific Editor: *Eric Sawey, PhD*

Transaction Report:

September 3, 2021

Re: Life Science Alliance manuscript #LSA-2021-01193-T

Dear Dr. Kwak,

Thank you for submitting your manuscript entitled "Testing of the therapeutic efficacy and safety of AMPA receptor RNA aptamers in an ALS mouse model" to Life Science Alliance. The manuscript was assessed by expert reviewers, whose comments are appended to this letter. We invite you to submit a revised manuscript addressing the Reviewer comments.

Thank you for this interesting contribution to Life Science Alliance. We are looking forward to receiving your revised manuscript.

Sincerely,

B. MANUSCRIPT ORGANIZATION AND FORMATTING:

Reviewer #1 (Comments to the Authors (Required)):

Manuscript entitled "Testing of the therapeutic efficacy and safety of AMPA receptor RNA aptamers in an ALS mouse model" by Akamatsu et al investigated the stability and distribution of FN1040 in vivo after intracerebroventricular infusion, and evaluated the efficacy by measuring changes in motor function, motor neuron numbers, and subcellular localization of TDP-43 in AR2 mice. Generally, the manuscript is well-organized and adds new knowledge to the current literatures. However, there are some concerns should be addressed.

Major concerns:

1. The authors wrote: a 12-week infusion of aptamers to AR2 mice has blocked the progression of motor dysfunction, normalized TDP-43 mislocalization, and prevented death of motor neurons. However, the authors did not describe the disease trajectory of AR2 mice, and 20-32 weeks stands which stage in the whole life of the mice. From Fig.4b, it seems that vehicle-treated AR2 mice didn't show any motor dysfunction during the 17-32 weeks observed. Therefore, it is difficult to say "blocked the progression of motor dysfunction". Meanwhile, death of motor neurons of AR2 mice were not shown in the results, non-transgenic mice should be used as control for both behavioral and pathological. From Fig. 3b, we can see that FN1040 only partly changed the mislocalization of TDP-43 compared to wild type mice, while WT mice was not listed in Fig.4f-g, the conclusion of normalization of TDP-43 mislocalization deserves further consideration.
2. Immunofluorescence images showing TDP-43 localization should be shown.
3. Fig.4c shows that FN1040 treated mice have similar body weight compared with vehicle-treated mice, indicating that body weight is not a variable for disease status?

Minor concerns:

1. Statistical term P should be used as capital, Italic font universally in the whole text. The statistical significance can be divided as two categories: * P <0.05, ** P < 0.01.

Reviewer #2 (Comments to the Authors (Required)):

In this manuscript, the authors show that FN1040 and FN58, two RNA aptamers, have protective effects in an ALS-associated in AR2 mouse model. Moreover, they show that administration of FN1040 in AR2 mice could rescue motor neurons in association with TDP-43 re-localization. Although this manuscript contains logical experimental design and interesting pharmacological experiments in animal model, it suffers from the lacking of mechanism study - since the use of small molecule AMPA receptor inhibitors in this AR2 mouse model have been studied, the use of RNA aptamers as a new type of AMPA receptor antagonists in the same model are not strong enough to support the publication in Life Science Alliance. The reviewer has the following comments and concerns:

1. The cellular localization of TDP-43 is the key experimental information in this manuscript, therefore the imaging data of TDP-43 need to be shown in the main figures. Also, the morphology analysis of ChAT-positive neurons need to be performed.
2. How does these AMPA receptor antagonists affect TDP-43 cellular localization? It regulates TDP-43 through AMPA receptor and calcium signaling? - can it directly target TDP-43?
3. The authors should test other biomarkers of Ca²⁺ influx beyond TDP-43 localization.
4. It would be helpful to compare the effect of FN1040 and FN58 with other known AMPA receptor inhibitors (small molecule compounds) in ALS mouse models.
5. The manuscript is overall well-written, but it still contains multiple writing errors.

Reviewer #3 (Comments to the Authors (Required)):

Akamatsu et al., investigated the effects of RNA aptamers targeting AMPA receptor on ALS phenotypes using AR2 mice. The group have reported that failure of GluA2 RNA editing resulting from downregulation of the RNA-editing enzyme ADAR2 occurs in the many of sporadic ALS cases and causes the death of motor neurons through AMPA receptor-mediated mechanism. AR2 mice, conditional ADAR2 knockout exhibited ALS phenotypes by AMPA receptor mediated mechanism. In this manuscript the authors showed that the continuous ICV infusion of RNA aptamers could rescue ALS phenotypes of AR2 mice. Two candidate

aptamers which were chemically modified with more prolonged stability in vivo could reverse motor function and ALS pathology in AR2 mice. The aptamers were administered at 5-month-old when AR2 mice started to exhibit ALS phenotypes. One of two candidate aptamers, FN1040 seemed to have less sedation side effects compared to FN58.

It is noteworthy that administration of RNA aptamers could reverse ALS-phenotype in AR2 mice regarding the therapeutic possibility. Overall, the work is well controlled. Some specific questions remain but should be addressable.

1. It would be desirable to show representative images of motor neurons in the spinal cord with the graphs, such as Fig. 3b-d, 4d,e, EV1a, and EV2a-b.
2. Some graphs are not statistically correct. For instance, Fig3b showed MN numbers from 18 anterior horn sections. But it was collated from three mice. Six anterior sections from the same mouse should have relevancy. The authors need to justify this issue and show in a different way. Probably same issues are in Fig3c, 4d, 4e, EV2a, and EV2b.
3. It would be better to include both FN58 and FN1040. It looks odd that Fig.1a-b showed the data of FN1040 only, while FN58 was solely used in Fig. 1c-d.
4. The measurements of Fig2c should be repeated and have error bars.
5. There was no statistical analysis in Fig.3a. Two-way repeated ANOVA may be applied to see the significant differences among groups. Otherwise, the authors should change the way to show the results.
6. Subcellular localization proportion graphs were not statistically analyzed at all. Number of samples and error bars should be required as well.
7. The time-course evaluation of rotarod and BW in Fig4b-c seemed statistically insufficient. Repeated ANOVA test is necessary. It would be recommended to ask a statistician.
8. It would be helpful to specify the disease onset of AR2 mice in the main text, even if they did not exhibit it in figures.

November 26, 2021

Dear all reviewers,

We carefully revised our manuscript in line with reviewers' comments as described below. All the changes are marked with blue words and underlines in the text, and we also attached the highlighted text as related manuscript files. revised manuscript is with 46 references, four figures, and ten supporting information tables and four figures.

Reviewer #1

Manuscript entitled "Testing of the therapeutic efficacy and safety of AMPA receptor RNA aptamers in an ALS mouse model" by Akamatsu et al investigated the stability and distribution of FN1040 in vivo after intracerebroventricular infusion, and evaluated the efficacy by measuring changes in motor function, motor neuron numbers, and subcellular localization of TDP-43 in AR2 mice. Generally, the manuscript is well-organized and adds new knowledge to the current literatures. However, there are some concerns should be addressed.

Major concerns:

1. The authors wrote: a 12-week infusion of aptamers to AR2 mice has blocked the progression of motor dysfunction, normalized TDP-43 mislocalization, and prevented death of motor neurons. However, the authors did not describe the disease trajectory of AR2 mice, and 20-32 weeks stands which stage in the whole life of the mice.

REPLY: First of all, we would like to thank the Reviewer for all their comments which helped us, in our opinion, to significantly improve the message of this manuscript. We have previously reported the disease trajectory of AR2 mice (Hideyama et al., The Journal of Neuroscience, September 8, 2010 • 30(36):11917–11925) as shown in the Fig. 1 and Fig. 2 for the reviewer below (or Fig. 3 and Fig. 4 in the original paper, respectively). The panels that show the time course of behavioral changes and the reduction in the number of anterior horn cells are each marked with a red box, indicating that disease is progressing during the age range we investigated in this study. To clarify the trajectory of AR2 mice, we therefore added the description about the behavior of AR2 mice at 20-32 weeks of age in page 10, line 2 in the result section as below indicated.

“We started the treatment of AR2 mice at 20 weeks of age when the motor neuron degeneration and motor dysfunction were clearly under way (Hideyama et al., 2010). Such a starting time was still considered an early phase of the disease progression. Unlike other ALS model mice such as SOD1 transgenic mice, the disease progression of AR2 mice is slow and represents more closely the time course of human ALS (Hideyama et al., 2010). As such, AR2 mice at 20 to 32 weeks of age are roughly in the middle stage of the disease progression.”

Fig. 1 for reviewers.

Figure 3. Behavioral changes in AR2 mice. *A*, Rotarod performance presented as latency to fall (at 10 rpm, 18 maximum) began to decline at 5 weeks of age in AR2 mice and rapidly fell to low levels during the initial 5–6 months, stable until 18 months of age. Control mice exhibited full performance (180 s) until ~12 months of age, followed by slight performance ($>164.5 \pm 6.4$ s) until 24 months. *B*, Grip strength measured declined with kinetics similar to those of performance. In *A* and *B*, the scores obtained for the AR2 mice (mean \pm SEM; $n = 28$) are indicated as percentage performance control mice ($n = 15$). *C*, AR2 mice exhibited slightly lower body weight than controls ($p > 0.05$). *D*, AR2 mice ($n = 33$) lifespans, but the rate of death increased after month 18. The median \pm SEM survival was 81.5 ± 16.4 weeks for compared with 105.1 ± 13.5 weeks for control mice ($p = 0.0262$, log-rank analysis).

Fig. 2 for the reviewers.

Figure 4. Loss of ADAR2-deficient motor neurons. *A*, Degenerating AHCs in AR2 mice at 2 months (2m; Nissl staining) and 4 months (4m; toluidine blue staining, 1 μ m section) of age. Scale bar: 2m, 25 μ m; 4m, 12.5 μ m. *B*, Ventral root (L5) of control (Ctl) and AR2 mice at 4 months of age (4m). Inset, Magnified view of degenerating axons. Scale bar: 100 μ m; inset, 20 μ m. *C*, Numbers of AHCs showing ADAR2 immunoreactivity (black columns) and lacking this immunoreactivity (gray columns) (mean \pm SEM) in AR2 mice at different ages (1m, 2m, 6m, 9m, 12m). In AR2 mice, Cre expression is developmentally regulated (orange line), and ~50% of motor neurons express Cre by 5 weeks of age, with recombination of the *ADAR2* gene in ~10% of AHCs at 1 month of age and 40–45% of AHCs after 2 months of age (orange line). The number of ADAR2-lacking AHCs significantly decreased in AR2 mice after 2 months of age as a result of Cre-dependent knock-out of *ADAR2* ($*p < 0.01$, repeated-measures ANOVA). The number of AHCs in the control mice did not change at different ages, and all the AHCs in controls showed ADAR2 immunoreactivity. *D*, Electrophysiological examination in AR2 mice. Electromyography from an AR2 mouse at 12 months of age showing fibrillations and fasciculations, common findings in ALS indicative of muscle fiber denervation and motor unit degeneration and regeneration.

From Fig.4b, it seems that vehicle-treated AR2 mice didn't show any motor dysfunction during the 17-32 weeks observed. Therefore, it is difficult to say "blocked the progression of motor dysfunction". Meanwhile, death of motor neurons of AR2 mice were not shown in the results, non-transgenic mice should be used as control for both behavioral and pathological.

REPLY: As the reviewer pointed it out correctly, the rotarod score of vehicle-treated mice may be best compared to that from our previous experiments (Fig. 3 of Hideyama et al., J Neurosci that is

presented in the response to your previous comment; Akamatsu et al, Sci Reports 2016, 6/28649). As such, the aptamer-treated mice displayed better performance as compared with the same group of the mice prior to aptamer-treatment. An increase of the rotarod performance could be counted for by the increase of the number of AHCs in the aptamer-treated mice, as compared with the vehicle-treated mice. We believe the reason of a lack of apparent deterioration of rotarod score is due to a slower disease progression during the time frame of our measurement. We also added the data on wild-type mice from the same strain as control in Fig. 4E, 4F and 4G (indicated as C57BL6wt). C57BL6 mice did not show any significant decline in the rotarod performance until 1 year of age (described previously as “Control mice exhibited full performance (180 s) until ~12 months of age, followed by slightly lower performance ($> 164.5 \pm 6.4$ s) until 24 months.” in page 11921 of the J Neurosci 2010 cited above).

We changed the word in the following sentence according to your comment and added the data on wild-type C57BL6 mice at 32 weeks as the pathological control in Fig. 4E~4G (indicated as C57BL6wt).

“A 12-week continuous, intracerebroventricular infusion of aptamers to AR2 mice has **blocked** the progression of motor dysfunction, normalized TDP-43 mislocalization, and prevented death of motor neurons.” (Abstract line 7)

was changed to

“A 12-week continuous, intracerebroventricular infusion of aptamers to AR2 mice has **reduced** the progression of motor dysfunction, normalized TDP-43 mislocalization, and prevented death of motor neurons.”

From Fig. 3b, we can see that FN1040 only partly changed the mislocalization of TDP-43 compared to wild type mice, while WT mice was not listed in Fig.4f-g, the conclusion of normalization of TDP-43 mislocalization deserves further consideration.

REPLY : TDP-43 localizes in the nucleoplasm in healthy cells in various organs (<https://www.proteinatlas.org/ENSG00000120948-TARDBP/tissue>). As shown in Figs. 3F and 4G, the majority of the AHCs exhibit TDP-43 in the nucleus in the wild type C57BL6 mice. As shown in Fig 3F and Fig 4G, an increase of the ratio of Nuc+Nuc/Cyto (blue column) with a concurrent decrease of the level of Cyto (yellow column) shows aptamer treatment was able to prevent TDP-43 from mislocalizing or forming aggregations in cytosol. In other words, aptamer treatment is capable of preventing TDP-43 nuclear depletion and thus its cytoplasmic accumulation. These changes may appear subtle, but together with an increase in the number of AHCs and improved rotarod performance, we believe that the increase of nuclear TDP-43 level in the aptamer-treated AR2 mice is the molecular basis for an increase in the number of AHCs and thus improved rotarod performance.

We have revised the description an TDP-43 mislocalization, as in the results section (page 11, line 4), to clarify this point.

“Although the improvement in normalizing TDP-43 mislocalization appeared to be subtle, as compared with the vehicle-treated mice, the rotarod functional test score showed a remarkable improvement, suggesting a significant proportion of motor neurons recovered their neuronal function after the aptamer treatment.”

2. Immunofluorescence images showing TDP-43 localization should be shown.

REPLY: Thank you for the appropriate comment. We added representative immunofluorescence images in Fig 3B and Fig. 4D.

3. Fig.4c shows that FN1040 treated mice have similar body weight compared with vehicle-treated mice, indicating that body weight is not a variable for disease status?

REPLY: We previously reported that AR2 mice exhibited body weight gain curve similar to wild-type mice till one year of age, and then slightly lower weight than wild-type mice throughout the life (Fig. 2 for the reviewers, panel C, adopted from Hideyama, 2010, J Neuroscience). If mice can live longer than 1.5 year, AR2 mice may have exhibited a reduction in the body weight, but they did not exhibit significant weight loss before they die at 1.5 year of age. In this study, we measured body weight to assess adverse effects, and a continuous infusion of FN1040 over the course of the treatment did not interfere body growth (or weight gain over the time course of aptamer treatment), indicating that FN1040 has no adverse effects affecting body weight.

We added description about our previous study on body weight as below (page 10, line 18)

“In fact, AR2 mice exhibited body weight gain over time, which was similar to control mice until about the age of one year (Hideyama et al., 2010). No overt disabilities were observed in either FN1040-treated or FN58-treated mice.”

Minor concerns:

1. Statistical term P should be used as capital, Italic font universally in the whole text. The statistical significance can be divided as two categories: * P <0.05, ** P < 0.01.

REPLY: We use capital, italic font for all “P”s in statistical terms in accordance with your comment.

Reviewer #2

In this manuscript, the authors show that FN1040 and FN58, two RNA aptamers, have protective effects in an ALS-associated in AR2 mouse model. Moreover, they show that administration of FN1040 in AR2 mice could rescue motor neurons in association with TDP-43 re-localization. Although this manuscript contains logical experimental design and interesting pharmacological experiments in animal model, it suffers from the lacking of mechanism study - since the use of small molecule AMPA receptor inhibitors in this AR2 mouse model have been studied, the use of RNA aptamers as a new type of AMPA receptor antagonists in the same model are not strong enough to support the publication in Life Science Alliance. The reviewer has the following comments and concerns:

REPLY: We respectfully disagree. As a control, we did include an FDA approved drug, perampanel (Fycompa), a small molecule compound and a known AMPA receptor antagonist, in a parallel experiment to the aptamer treatment. As shown in Fig 3A, perampanel treatment caused a significant sedation in AR2 mice at a dose that was therapeutically efficacious. Furthermore (and this is also relevant to the question of body weight the first reviewer asked), the perampanel-treated AR2 mice failed to gain body weight (we have previously published a comprehensive study of administering perampanel in AR2 mice as in Akamatsu et al, Sci Reports 2016, 6/28649). As we described, a poor water solubility and known off-target activity of this small molecule compound could be the culprit of these significant side effects. However, aptamer FN1040, which selects AMPA receptors as a potent antagonist, was shown in the current study that it was both efficacious in rescuing motor neurons, normalizing TDP-43 mislocalization and improving motor function of the AR2 mice, yet without any detectable side effects. At the mechanistic level, we characterized FN1040 against each of the glutamate receptor channels using HEK-293 cells that expressed one receptor at a time. And, our preliminary data further showed that, by the use of two-photo microscopy, FN1040 was capable of blocking Ca^{2+} influx, based on the real time monitoring of 15 live cortical neurons from a live AR2 mouse. Therefore, we believe our study was mechanistically driven.

1. The cellular localization of TDP-43 is the key experimental information in this manuscript, therefore the imaging data of TDP-43 need to be shown in the main figures. Also, the morphology analysis of ChAT-positive neurons need to be performed.

REPLY: We have added the representative imaging data of TDP-43 in Fig 3B (typical localization pattern of TDP-43), Fig 3C (2-week administration) and Fig 4D (12-week administration), where the morphological features of the cells can be seen.

2. How does these AMPA receptor antagonists affect TDP-43 cellular localization? It regulates

TDP-43 through AMPA receptor and calcium signaling? - can it directly target TDP-43?

REPLY: We have reported how exaggerated Ca^{2+} influx through abnormal AMPA receptor activities affects cellular TDP-43 localization (Yamashita et al., Nature Communications 3:1307, 2012). We found that the expression of Q/R site-unedited GluA2 or GluA2Q isoform increases in the motor neurons in which the expression of ADAR2 is either reduced or absent, as in ALS patients and in the AR2 mouse model we used in this study. The Q/R site-unedited GluA2 or GluA2Q can self-assemble into a functional channel that is abnormally Ca^{2+} -permeable. The abnormal activity of GluA2Q AMPA receptors leads to the activation of calpain (Ca^{2+} -dependent cysteine protease), which then serially cleaves TDP-43 from its C-terminal. The fragmented TDP-43 becomes aggregated when calpain is continuously activated by Ca^{2+} influx through the AMPA receptors. As TDP-43 localized mainly in the nucleus but shuttles between the nucleus and the cytoplasm, the whole length TDP-43 from the nucleus is also trapped to the aggregates of fragment TDP-43 in the cytoplasm. We have demonstrated that mice able to overexpress Q/R site-edited GluA2 or GluA2R in spite of the lack of ADAR2 in the motor neurons (AR2res mice) exhibited normal nuclear localization of TDP-43. Therefore, blocking the excess influx of Ca^{2+} into neurons by AMPA receptor antagonists prevents mislocalization of TDP-43 and death of motor neurons. Not surprisingly, using AMPA receptor antagonists such as perampanel (i.e., 2-(2-oxo-1-phenyl-5-pyridin-2-ylpyridin-3-yl)benzotrile) can prevent TDP-43 mislocalization and aggregation in the cytoplasm, as we showed in a previous study (Akamatsu et al, Sci Reports 2016, 6/28649). We have provided a brief summary about the link of RNA editing, AMPA receptor Ca^{2+} permeability with TDP-43 pathology on page 3 of the Introduction section in the manuscript.

We should note that the RNA aptamers we tested in this current study have been designed to target AMPA receptors and inhibit these receptor activities. Like we described above and shown in this current study, blocking abnormal AMPA receptor activities is sufficient to inhibit TDP-43 pathology.

Incidentally, blocking abnormal AMPA receptor activity using AMPA receptor antagonists like our aptamers was intended to mechanistically inhibit the abnormal Ca^{2+} influx through abnormally expressed AMPA receptors, thus preventing motor neuron degeneration by the so-called excitotoxicity pathway. At the molecular level, such an approach was to also block the TDP-43 pathology via Ca^{2+} -dependent cysteine protease activation, which is initiated through AMPA receptor-mediated Ca^{2+} influx. Therefore, the design of this experimental work such as from the design of the aptamers to the use of the AR2 mice was again entirely mechanism driven.

3. The authors should test other biomarkers of Ca^{2+} influx beyond TDP-43 localization.

REPLY: First of all, there is no biomarker or test of disease activity for ALS. That said, cytoplasmic mislocalization and aggregation of TDP-43 are found in 98% of all cases of ALS (in both sporadic and familial ALS cases). Thus, TDP-43 pathology is considered the hallmark of ALS. Regardless of what biomarker(s) and a disease mechanism(s) one studies, in the end, one key outcome is to rescue or prevent selective motor neuron death. Second, Ca^{2+} influx through the Ca^{2+} -permeable GluA2Q AMPA receptors have been extensively investigated by many different research groups during and after 1990s. Specific roles of Ca^{2+} -permeable AMPA receptors in causing the death of motor neurons have been well established (e.g. Kwak & Weiss, *Curr Opin Neurobiol* 16:281-7, 2006). We have previously reported (Hideyama et al, *J Neurosci* 2010; Yamashita et al, *Nat Commun* 2012; Akamatsu et al., *Sci Reports* 2016), and again shown in the current study, that blockade of Ca^{2+} influx through these abnormal AMPA receptors effectively rescues motor neuron death and improves motor neuron health in terms of increasing both the number and the size of motor neurons. We have also demonstrated that subcellular TDP-43 localization (i.e., the ratio of the TDP-43 fraction in nucleus to that in cytoplasm) is a good biomarker for toxic Ca^{2+} influx build-up mediated through the AMPA receptors. An improvement of subcellular TDP-43 localization (Fig. 3F and 4G) is directly linked to an improvement of motor activities of the AR2 mice. We hope the reviewer will accept that the monitoring TDP-43 mislocalization is a critical, reliable and key biochemical indicator for the motor neuron health. Third, if we use a broad range of measurements as different ways to assess the outcome of our study, we have indeed tested, in addition to TDP-43 localization, a basket of parameters, such as motor neuron health (both in number and in size), behavioral measurements (rotarod experiments and mobility measurements) as well as body weight. Thus, our conclusion that FN1040 is efficacious has been based on this broad range of outcome measures, rather than a single TDP-43 measurement. It should be emphasized that FN1040 was not designed as a diagnostic tool molecule; rather it was designed as an antagonist to block motor neuron death due to AMPA receptor-mediated excitotoxicity. For this purpose, we believe FN1040 has done what it was designed to do.

4. It would be helpful to compare the effect of FN1040 and FN58 with other known AMPA receptor inhibitors (small molecule compounds) in ALS mouse models.

REPLY: We have previously studied, perampanel, an AMPA receptor antagonist, in the same AR2 mouse model, and the results have already been published (Akamatsu et. al., *Sci. Report* 6:28649, 2016). Perampanel was found to improve behavioral, morphological and immunohistochemical changes. Like we previously described (Akamatsu et. al., *Sci. Report* 6:28649, 2016) and showed in a parallel experiment in the current study where a mouse group was randomly chosen, we observed the sedative effect from the AR2 mice that were given perampanel. In contrast, we observed no significant sedative effect when FN1040 was continuously infused into the AR2 mice, as compared

with the vehicle control. Like we have described earlier, perampanel is a known noncompetitive AMPA receptor antagonist (Hanada et al. *Epilepsy* 52:1331-1340, 2011) and an FDA approved drug for treatment for partial onset seizures. Thus, the answer to the question for this reviewer is yes, we have done the experiments with a known small-molecule inhibitor of AMPA receptors, and we have concluded, as in the text, that FN1040 was superior to perampanel because the use of RNA aptamer produced efficacy but generated no sedative or any other adverse effects.

5. The manuscript is overall well-written, but it still contains multiple writing errors.

REPLY: We carefully corrected the errors.

Reviewer #3

Akamatsu et al., investigated the effects of RNA aptamers targeting AMPA receptor on ALS phenotypes using AR2 mice. The group have reported that failure of GluA2 RNA editing resulting from downregulation of the RNA-editing enzyme ADAR2 occurs in the many of sporadic ALS cases and causes the death of motor neurons through AMPA receptor-mediated mechanism. AR2 mice, conditional ADAR2 knockout exhibited ALS phenotypes by AMPA receptor mediated mechanism. In this manuscript the authors showed that the continuous ICV infusion of RNA aptamers could rescue ALS phenotypes of AR2 mice. Two candidate aptamers which were chemically modified with more prolonged stability in vivo could reverse motor function and ALS pathology in AR2 mice. The aptamers were administered at 5-month-old when AR2 mice started to exhibit ALS phenotypes. One of two candidate aptamers, FN1040 seemed to have less sedation side effects compared to FN58.

It is noteworthy that administration of RNA aptamers could reverse ALS-phenotype in AR2 mice regarding the therapeutic possibility. Overall, the work is well controlled. Some specific questions remain but should be addressable.

1. It would be desirable to show representative images of motor neurons in the spinal cord with the graphs, such as Fig. 3b-d, 4d,e, EV1a, and EV2a-b.

REPLY: We thank the Reviewer for these suggestions that have been addressed. We have added representative immunohistochemical images of the motor neurons to Fig. 3B, 3C and Fig. 4D.

2. Some graphs are not statistically correct. For instance, Fig3b showed MN numbers from 18 anterior horn sections. But it was collated from three mice. Six anterior sections from the same mouse should have relevancy. The authors need to justify this issue and show in a different way.

Probably same issues are in Fig3c, 4d, 4e, EV2a, and EV2b.

REPLY: In line with the comment, we re-analyze the data by comparing the mean of each mouse (Figs. 3D, 3E and 4E-4G), and moved the data by comparing the mean of the number in the anterior horn to supplementary data. The results in the number of AHCs were not significantly different between mouse-based analysis and AH-based analysis. Only the dose-dependency of the ChAT-positive AHCs did not reach statistical significance by mouse-based analysis, although the difference between non-treated group and the groups treated with aptamers was statistically significant. This is probably due to the small number of mice in each group. We changed the relevant part in the text (page 8, line 10) and the legend for relevant figures.

“In fact, the increase of the number of ChAT positive motor neurons was dose-dependent, and the increase was the highest in mice treated with 20 μ M of FN1040 (Fig 3D. and Fig S2B.). The size of ChAT-positive neurons in mice that received FN1040 was also larger as compared to vehicle-treated mice (Fig 3E. and Fig S2C.).”

3. It would be better to include both FN58 and FN1040. It looks odd that Fig.1a-b showed the data of FN1040 only, while FN58 was solely used in Fig. 1c-d.

REPLY: Agreed. However, we hope it is acceptable to the reviewer if we include the FN1040 data in supplemental figure 1 (Fig S1), since the data on FN1040 has already been previously published (Huang et al ACS Chem Neurosci. 8:2437-2445, 2017).

4. The measurements of Fig2c should be repeated and have error bars.

REPLY: Accepted the recommendation. We revised Fig 2C to illustrate the repeated data with error bars.

5. There was no statistical analysis in Fig.3a. Two-way repeated ANOVA may be applied to see the significant differences among groups. Otherwise, the authors should change the way to show the results.

REPLY: According to the comment, we increased the number of mice to n=5 and performed statistical analysis in Fig 3A by two-way ANOVA.

6. Subcellular localization proportion graphs were not statistically analyzed at all. Number of samples and error bars should be required as well.

REPLY: We analyzed the data and provided the number of samples and error bar in all relevant graphs (Fig. 3F and Fig. 4G).

7. The time-course evaluation of rotarod and BW in Fig4b-c seemed statistically insufficient. Repeated ANOVA test is necessary. It would be recommended to ask a statistician.

REPLY: We accepted the recommendation. Specifically, we consulted to the statistician and performed a two-way ANOVA followed by Tukey's multiple comparison test on the time-course study in Fig. 4B and 4C.

8. It would be helpful to specify the disease onset of AR2 mice in the main text, even if they did not exhibit it in figures.

REPLY: As described in the reply to the first comment by Reviewer #1, we added figures from our previous study as Fig. 1 for reviewers in this letter to help our answering the questions. The time-course of the phenotype in this AR2 mouse model is a slow one, similar to ALS. Motor neurons die after surviving in a dysfunctional state for a variety of time periods, and behavioral change will appear only after certain proportions of motor neurons die in AR2 mice and ALS patients as well. Therefore, like in clinical ALS diagnostics, it is difficult to specify the time of initiation of disease onset in the AR2 mouse model. That said, we added a description of the disease onset of AR2 mice in the results (page 10, line 2).

“We started the treatment of AR2 mice at 20 weeks of age when the motor neuron degeneration and motor dysfunction were clearly under way (Hideyama et al., 2010). Such a starting time was still considered an early phase of the disease progression. Unlike other ALS model mice such as SOD1 transgenic mice, the disease progression of AR2 mice is slow and represents more closely the time course of human ALS (Hideyama et al., 2010). As such, AR2 mice at 20 to 32 weeks of age are roughly in the middle stage of the disease progression.”

Sincerely yours,

Shin Kwak, MD, PhD

December 15, 2021

RE: Life Science Alliance Manuscript #LSA-2021-01193-TR

Dr. Shin Kwak
Tokyo Medical University
Neurology
6-7-1 Nishi-shinjuku
Shinjuku-ku, Tokyo 160-0023
Japan

Dear Dr. Kwak,

Thank you for submitting your revised manuscript entitled "Testing of the therapeutic efficacy and safety of AMPA receptor RNA aptamers in an ALS mouse model". We would be happy to publish your paper in Life Science Alliance pending final revisions necessary to meet our formatting guidelines.

- please add ORCID ID for the corresponding (and secondary corresponding) author-you should have received instructions on how to do so
- please add the Twitter handle of your host institute/organization as well as your own or/and one of the authors in our system
- please note that the titles in the system and the manuscript file must match
- please consult our manuscript preparation guidelines <https://www.life-science-alliance.org/manuscript-prep> and make sure your manuscript sections are in the correct order
- please add your main, supplementary figure, and table legends to the main manuscript text after the references section
- please use the [10 author names, et al.] format in your references (i.e. limit the author names to the first 10)
- please add callouts for Figure S1A, B to your main manuscript text

FIGURE CHECKS:

- please include scale bars on Figures 2B and 3B, and indicate their size in the legend

A. FINAL FILES:

B. MANUSCRIPT ORGANIZATION AND FORMATTING:

Sincerely,

Reviewer #1 (Comments to the Authors (Required)):

Satisfied. No further comments.

Reviewer #3 (Comments to the Authors (Required)):

The manuscript is improved and now suitable for publication by addressing most of issues raised in the 1st review.

December 23, 2021

RE: Life Science Alliance Manuscript #LSA-2021-01193-TRR

Dr. Shin Kwak
Tokyo Medical University
Neurology
6-7-1 Nishi-shinjuku
Shinjuku-ku, Tokyo 160-0023
Japan

Dear Dr. Kwak,

Thank you for submitting your Research Article entitled "Testing of the therapeutic efficacy and safety of AMPA receptor RNA aptamers in an ALS mouse model". It is a pleasure to let you know that your manuscript is now accepted for publication in Life Science Alliance. Congratulations on this interesting work.

DISTRIBUTION OF MATERIALS:

Again, congratulations on a very nice paper. I hope you found the review process to be constructive and are pleased with how the manuscript was handled editorially. We look forward to future exciting submissions from your lab.

Sincerely,
